# Influence of Graphene Oxide on Mechanical and Morphological Properties of Nafion^®^ Membranes

**DOI:** 10.3390/nano15010068

**Published:** 2025-01-03

**Authors:** Carlos Ceballos-Alvarez, Maziar Jafari, Mohamed Siaj, Samaneh Shahgaldi, Ricardo Izquierdo

**Affiliations:** 1Département de Génie Électrique, École de Technologie Supérieure, 1100 Notre-Dame Street West, Montreal, QC H3C 1K3, Canada; juan-carlos.ceballos-alvarez.1@ens.etsmtl.ca; 2Département de Chimie, Université du Québec à Montréal, 2101 Rue Jeanne-Mance, Montreal, QC H2X 2J6, Canada; jafari.maziar@courrier.uqam.ca (M.J.); mohamed.siaj@usherbrooke.ca (M.S.); 3Institute de Recherche sur l’Hydrogene, Université du Québec à Trois-Rivières, 3351, Boul. des Forges C.P. 500, Trois-Rivières, QC G9A 5H7, Canada; samaneh.shahgaldi@uqtr.ca

**Keywords:** graphene oxide, Nafion^®^, PEM water electrolysis, ultrasonic spray, mechanical properties

## Abstract

This study explored the influence of graphene oxide (GO) on morphological and mechanical properties of Nafion^®^ 115 membranes with the objective of enhancing the mechanical properties of the most widely employed membrane in Proton Exchange Membrane Water Electrolyzers (PEMWE) applications. The membrane surface was modified by ultrasonically spraying a GO solution and different annealing temperatures were tested. Scanning Electron Microscopy (SEM) cross-sectional images revealed that annealing the composite membranes was sufficient to favor an interaction between the graphene oxide and the surface of the Nafion^®^ membranes. The GO covering only 35% of the membrane surface increased the composite’s wettability from hydrophobic (105.2°) to a highly hydrophilic angle (84.4°) while slightly reducing membrane swelling. Tensile tests depicted an increase in both the strain levels and tensile loads before breaking. The samples with GO presented remarkable mechanical properties when the annealing time and temperature increased; while the Nafion^®^ control samples failed at elongations of 95% and 98%, their counterparts with GO on the surface achieved elongations of 248% and 191% when annealed at 80 °C and 110 °C respectively, demonstrating that the presence of GO mechanically stabilizes the membranes under tension. In exchange, the presence of GO altered the smoothness of the membrane surface going from an average 1.4 nm before the printing to values ranging from 8.4 to 10.2 nm depending on the annealing conditions which could affect the quality of the subsequent catalyst layer printing. Overall, the polymer’s electrical insulation was unaffected, making the Nafion^®^-GO blend a more robust material than those traditionally used.

## 1. Introduction

Recently, the use of hydrogen as a way to decarbonize our energy systems has gained considerable attention worldwide with several countries even launching their hydrogen national strategies [1,2,3,4]. The idea of chemically storing excess intermittent renewable energies in an energy carrier such as hydrogen has attracted considerable attention [5,6,7]. Among the known technologies to produce hydrogen, proton exchange membrane water electrolysis (PEMWE)offers several appealing industrial advantages over traditional methods, including greater energy efficiency, higher production rates due to faster electrode kinetics, a more compact design, enhanced safety, and dynamic operation [8,9,10,11].

Perfluorosulfonic acid (PFSA) polymer membranes are widely used as the solid electrolyte in both Proton Exchange Membrane (PEM) water electrolyzers and fuel cells [12]. The most commonly used and commercially successful PEMs are Nafion^®^ PFSA membranes [13,14], which are used as a benchmark due to their excellent mechanical and thermal stability, high proton conductivity, and swelling behavior [15,16,17,18,19,20]. However, during operation, chemical degradation and mechanical failure due to hydration and temperature cycles are the most prominent factors restricting the durability of Nafion^®^ PEMs [21,22,23,24]. Moreover, in real-world applications, the degradation through thinning or the development of pinholes proved to accelerate complete system failure [25].

Fabricating composites has been attempted to improve Nafion^®^ mechanical resistance. The use of carbon-based nanomaterials such as multiwalled carbon nanotubes (MWCNT) [26,27,28] graphene [20,29,30,31] and Graphene Oxide (GO) [32,33,34,35] has been extensively studied. Conversely, while MWCNT and graphene are attractive to reinforce the membranes mechanically, their high electrical conductivity can cause electronic crossover through the membrane [28] and lower fuel cell voltages [36]. GO prevails over graphene derivatives as a Nafion^®^ reinforcing material due to its electric insulating properties [37,38], chemical stability, large specific surface area, high ion carrier mobility, mechanical strength [39], and the ability to form stable aqueous colloids by simple and cheap solution processes [37]. Moreover, GO’s presence of its multiple organic functional groups can act as chemical grafting sites for the hydrophilic sulfonic acid groups (-SO_3_H) in Nafion^®^. Recent studies have reported that these functional groups enhance the conductive performance of the resulting nanocomposite membranes while significantly improving their mechanical, thermal, and chemical resistance [40,41,42].

GO offers significant advantages over other nanocarbons, such as graphene and MWCNTs, when incorporated into Nafion^®^. Its oxygen-containing functional groups enhance mechanical stability and improve hydrophilicity, favoring the water uptake crucial for proton conductivity. Unlike graphene and MWCNTs, which are hydrophobic due to their non-polar, sp^2^-bonded carbon surfaces, the hydrophilic nature of GO makes it more compatible with the polymer matrix. Additionally, the lower electrical conductivity of GO minimizes parasitic conduction, preserving electrochemical efficiency. A comparison in performance of some of these nanocarbons can be seen in Appendix A [34,42,43,44].

In the previously mentioned works, solution casting have been the most sought after method to produce the nanocarbon–Nafion^®^ composite membranes. However, this process is labor intensive, time-consuming, and is not roll-to-roll compatible. Hence, it is less suitable for an industry setting.

Interestingly, the ultrasonic spraying technique has been reported to exfoliate GO layers [45,46]. Using this method, the material dispersion is ultrasonicated both in the liquid container and in the nozzle itself while the ink is being printed. Thus, we can prevent agglomeration and deliver a uniform distribution [25] with droplet sizes that can be controlled by adjusting the nozzle frequency to obtain very reproducible results [45]. Unlike conventional spray coating and solution casting methods, this technique can be easily automated and thus is more amenable to large-scale industrialization [47].

The objective of this work is to modify the Nafion^®^ 115 membrane surface by ultrasonic spraying of GO, to produce a novel GO–Nafion^®^115 composite membrane. The latter is comprehensively characterized in the present article. The morphological and mechanical properties of the composite membrane subjected to different annealing temperatures is reported. The influence of GO on the membrane morphology and wettability was investigated by AFM, SEM and contact angle measurements, while assuring electrical insulation on the composite membranes. The local nanomechanical mapping, bulk tensile tests, DMT/Young’s moduli, the breaking strength, and the elongation percentage were evaluated and compared. To the best of our knowledge, this is the first study to inquire the effects of such pretreatment conditions.

## 2. Materials and Methods

### 2.1. Graphene Oxide Synthesis

GO was synthesized from natural graphite by the modified Hummers method [48] and as detailed by Brisebois et al. [49]. Deionized water was used as a solvent to produce a GO ink with a concentration of 1 mg/mL. The formulated ink showed long-term stability and excellent dispersion in water [50,51] even months after the ink formulation, due to the presence of hydrophilic functional groups [52].

### 2.2. Ultrasonic Spray

Nafion^®^ 115 membranes with a thickness of 127 microns were purchased from Fuel Cell Store (Bryan, TX, USA) and were used as received. A printed area of 20 mm × 20 mm was used per sample. The GO ink was sonicated for 3 h then ultrasonically sprayed directly onto the Nafion^®^ 115 membrane using a Sono-Tek ‘Exacta-Coat’ ultrasonic spray instrument (Milton, NY, USA) vibrating at a high frequency of 48 kHz that produced a GO loading of 0.125 mgGO/cm^2^. Raster scans offset by 0.5 mm ensured homogeneous coverage. An ink flow rate of 0.1 mL/min, nozzle power of 1.1 watts, an air flow of 2 L/min and a fixed nozzle-to-substrate distance of 6.1 cm were configured for all samples. Control samples of Nafion^®^ 115 membranes were prepared by spraying only water. The membranes were left to dry and stabilize for 24 h at room temperature in a clean room before testing.

### 2.3. Thermal Treatment

To favor an interaction between the deposited GO and the Nafion^®^ polymer, heat treatments were performed. In PEMWE, the presence of the polymeric membrane typically limits the electrolysis temperatures to 80 °C [53]. The composite membranes were thermal treated in ambient air at this temperature for 3 and 12 h to partially emulate the membrane’s operative conditions. Additional annealing was performed at the polymer’s low glass transition temperature (110 °C) for 3 h to explore the influence of the presence of GO on the polymer’s mechanical properties in conditions where its original strength was slightly degraded.

### 2.4. Characterization

A comprehensive set of characterization methods was employed to evaluate the physical, mechanical, and surface properties of the Nafion^®^-GO composite membranes. These analyses provided valuable insights into the surface morphology, structural integrity, wettability, mechanical strength, dimensional stability, and electrical properties. Detailed descriptions of each technique and their results are presented in the following sections.

#### 2.4.1. Atomic Force Microscopy

To characterize the surface morphology of the membranes, Nafion^®^ 115 and Nafion^®^ 115/GO film samples were sliced and fixed on a 12 mm diameter magnetic disc with double-sided carbon tape. An extensive study of topography and quantitative nanomechanical (QNM) features was conducted using a Multimode8 (Bruker, Billerica, MA, USA) AFM in QNM peak-force mode, equipped with a 125 μm × 125 μm limit scanner and a ${\mathrm{Si}}_{3}N_{4}$ cantilever tip in tapping mode. The tip deflection sensitivity was calibrated by scanning a standard sapphire sample, and the tip radius of curvature was measured to be 4 nm by scanning a standard tip-check sample followed by image processing using the Tip Qualification tool in the Nanoscope Analysis (v1.40r1) image processing software. The cantilever resonant frequency and force constant were determined to be 128 kHz and $4.8\;N\cdot m^{-1}$, respectively, by the thermal tuning method. The QNM peak-force scan mode enabled high-resolution local mapping of the composite membrane’s elastic modulus using the Derjaguin-Muller-Toporov (DMT) model. Images were compiled of 512 raster scanned lines at a rate of 1 line/s, returning a quality of 512 × 512 pixels. The images were processed with the free open-access software Gwyddion (v3.0) or WSxM (v5.0 Develop 10.3) [54].

#### 2.4.2. Scanning Electron Microscopy

The morphology of the composite membranes was investigated using a high-resolution scanning electron microscope (Hitachi, SU-8230 FE-SEM, Chiyoda, Japan) equipped with a Bruker QUANTAX FlatQUAD detector. For SEM imaging, samples were coated with 4 nm of platinum in a vacuumed chamber using a turbo pumped sputter coater/carbon coater (Q150T, Guelph, ON, Canada).

#### 2.4.3. Contact Angle Measurements

Contact angle measurements were conducted using a Dataphysics OCA-20 goniometer (DataPhysics Instruments, USA) to quantify the wettability of the composite Nafion^®^-GO membranes. DI water droplets of 5 μL were dispensed on the surface, and all the angles were measured once the membrane was stabilized; images were analyzed using a SCA20 6.1 video-based measurement of static contact angles software.

#### 2.4.4. Tensile Tests

Tensile tests were conducted with an MTS AllianceTM RF/200 material testing system (USA). Samples were cut and tested according to ASTM Standards D6287 [55] and D882 [56], respectively. Membrane specimens of identical dimensions (150 mm × 10 mm) were employed in all cases for comparison. Measures were performed in triplicate.

#### 2.4.5. Membrane Swelling

Membrane water swelling was measured in triplicate on samples with dimensions of 10 mm × 10 mm following the procedure reported elsewhere [14]; briefly, samples were first dried in an oven at 60 °C for two hours, and the in-plane geometrical dimension was then immediately determined using a micrometer (Mitutoyo, ±1 μm). Samples were then placed into a water bath at room temperature for one hour. After saturation, excess surface water was removed carefully with tissue paper, and their geometrical expansions were immediately measured. Swelling (Sw) was calculated by using Equation (Equation 1), where $T_{\mathrm{wet}}$ and $T_{\mathrm{dry}}$ are the wet and dry dimensions of the samples, respectively:(1)$$Sw\left(\%\right)=\frac{T_{\mathrm{wet}}-T_{\mathrm{dry}}}{T_{\mathrm{dry}}}\times 100$$

#### 2.4.6. Electrical Characterization

Since the GO was only deposited on the surface of the polymer, electrical measurements of the composite membranes were conducted on the GO printed side using a four-point probe setup. An S-302 instrument with a pin distance of approximately 1 mm was used at a temperature of 25 °C. Three measurements were performed for each sample.

Voltage–current (V-I) curves were recorded at three different points on each membrane to confirm uniformity. The sheet resistance ($R_{s}$) and electrical conductivity (σ) were calculated as described in Equations (Equation 2) and (Equation 3), respectively, with the geometric correction factor (*k*) applied as per the probe configuration.

All measurements were carried out at ambient temperature, ensuring consistent conditions for direct comparison of the modified and unmodified membranes’ performance

## 3. Results and Discussion

### 3.1. Morphology

In this section, we analyze and describe the membranes’ physical structure and surface features, focusing on their microscopic and nanoscopic characteristics. This will provide detailed insights into the shape, size, distribution, and organization of the components that form the membrane.

#### 3.1.1. Composite Membrane Morphology

The conventional PFSA polymer electrolyte limits the operating temperature of PEM water electrolysis to 80 °C. Beyond this temperature, dehydration, loss in proton conductivity and increased degradation occur [57,58]. The GO–Nafion^®^ composite membranes were hence annealed at a working temperature of 80 °C for 3 and 12 h, and an additional test was performed at 110 °C for 3 h to explore the membrane mechanical behavior when close to its low glass transition temperature (110 °C) [59,60,61,62]. As the annealing time and temperature increased, the composite membrane changed gradually from transparent to a brownish color (Appendix A), a result of the molecular rearrangement during the polymer relaxation process [63].

The selected ultrasonic printing parameters resulted in a GO loading of 0.125 mgGO/cm^2^ producing GO islands of different sizes but regularly distributed on the surface of the Nafion^®^ membrane (Figure 1a). An image analysis of several samples shows a GO surface coverage between 34 and 36%. Higher magnifications of the islands show wrinkled GO sheets (Figure 1b,c). A similar wrinkled surface morphology was reported by others when printing GO [64,65,66].

GO islands formed on the surface of the PFSA membrane as a consequence of the solvent slowly evaporating at room temperature following ultrasonic spraying. During evaporation, capillary forces within the GO–water dispersion cause the GO sheets to compress, resulting in characteristic wrinkling. These forces pull the GO sheet together as the solvent recedes, often leading to deformation as the GO platelets settle into compact, lower-energy configurations.

#### 3.1.2. Roughness and Topography

An AFM topography analysis of different GO islands deposited on the surface of the Nafion^®^ 115 membrane was consistent with the SEM images, also showing a wrinkled morphology (Figure 1d). The height of the islands ranged between 7.5 and 9 nm at the interface, while zones in the center of the islands reached heights of 46 and up to 120 nm. The thinner GO islands near the Nafion^®^ interface resulted from the slow evaporation of the solvent at ambient temperature, a behavior typical of low-viscosity inks with a significant amount of solvent [67].

The Root Mean Square (RMS) roughness of the samples was analyzed by AFM. The GO islands presented average roughness values of 14.37 nm (Appendix A), while the surface of the Nafion^®^ membranes presented a considerably lower RMS roughness of 1.21 nm. Due to some vertically aligned flake protrusions, structural deformation, and the presence of covalently bonded functional groups [50,68,69], the GO sheets increased the roughness of the composite membranes. Appendix A shows the GO RMS roughness under the different annealing conditions. GO islands roughness values markedly decreased when the sample was treated for both long periods of time and at higher temperatures, with RMS values of 8.4 and 10.2 nm, respectively. This suggests that prolonged thermal treatment of the composite mebranes led to a height decrease in the surface wrinkles typically associated with GO, resulting in a smoother composite membrane surface.

Cross-sectional SEM imaging of the GO deposited on the surface of the PFSA membranes revealed stacked configurations consistent with the multilayered structures (Figure 2; additional images are provided in Appendix A). The GO layers exhibited characteristic surface wrinkles, a phenomenon attributed to van der Waals forces between the nanosheets [70]. The thickness of these sheets was measured to be 11–12 nm, and the interaction between Nafion^®^ and GO in non-annealed samples appeared more detached from the Nafion^®^ surface (Figure 2a).

In contrast, the annealed samples exhibited a more intimate binding at the GO–Nafion^®^ interface (Figure 2b,c). The images revealed that the annealing of the membranes promoted stronger connections between the two materials. This supports prior research suggesting that the absence of thermal treatment can lead to inadequate interfacial adhesion between GO and the polymer matrix, limiting the benefits of nanohybrid PEMs [71].

#### 3.1.3. Adhesion of the GO to the Surface of Nafion^®^ Membranes

The study investigated how the presence of GO affects the wettability of the membranes. The influence of annealing the composite membranes was evaluated with contact angle tests. When a 5 μL drop was placed on the surface of the non-annealed membranes, the drop dissolved the GO, leaving a void polymer surface (Appendix A). In contrast, the annealed membranes remained unchanged after the test. Consistent with the earlier SEM observations, this suggests that the thermal treatment facilitated stronger bonding between the GO and the Nafion^®^ interface.

This attachment effectively explains the observed GO water resistance observed on Appendix A. The rough and wavy surface topography of graphene oxide allows for mechanical interlocking with the polymer chains [72]. Its excellent compatibility and strong interfacial attraction with the polymer indicated potential mechanical improvement [73].

The connections formed between the ultrasonic sprayed GO and the polymer are noteworthy because the wrinkled configuration of the graphene oxide platelets are highly effective in suppressing crack propagation in polymer materials [72]. Figure 2d shows the interface between a GO island and the polymer membrane showing several cracks on the membrane surface.

These cracks result from temperature changes, humidity, and swelling during the printing and annealing processes [74]. Interestingly, the cracks disappeared in proximity to the GO islands. To investigate whether the GO–Nafion^®^ interface contributes to preventing the propagation of these defects through the chemical grafting of GO and the hydrophilic sulfonic acid groups (-SO_3_H) present in Nafion^®^, and to evaluate the enhancement of the mechanical properties of the composite membranes, tensile tests were conducted.

#### 3.1.4. Wettability of the Composite Membranes

Due to the proton permeability through water molecules existing within the membrane [75], water content, wettability, and the swelling of the membranes are critical factors in PEMWE systems. Contact angle measurements were performed to clarify how GO alters the wettability of the polymer. Figure 3 summarizes the measured values of the samples. The results indicate that the contact angle was consistently lower for samples with GO present on the surface compared to the Nafion^®^ control samples regardless of the experimental conditions. The presence of GO significantly increases wettability, shifting the contact angle from hydrophobic to hydrophilic [76]. This phenomenon is attributed to the oxygen-containing functional groups on the surface of GO including the carboxylic, hydroxyl, epoxy and carbonyl groups [77,78].

Improving membrane hydrophilicity is desirable, as the conductivity of a dry membrane is significantly lower—by several orders of magnitude—than that of a fully saturated one. Proton conductivity increases exponentially with water activity in the membrane [59] and can only reach optimal levels when the membrane is fully hydrated [79]. Therefore, incorporating GO to enhance the membrane’s wettability facilitates better water penetration into the polymer, promoting higher conductivity. Increased water availability within the membrane is essential for maintaining optimal proton conductivity, especially under high current densities and operational conditions, where dehydration risks are prevalent, and better hydration reduces the likelihood of localized dry spots, which could compromise the membrane’s mechanical and electrochemical performance.

### 3.2. Mechanical Characterization

This section focuses on the improvements in mechanical properties achieved for the Nafion^®^ membrane following GO functionalization. To assess the effects of this modification, a combination of tensile tests, nanoindentation, and nanomechanical mapping were employed. These techniques were used to evaluate how the proposed modifications influence the mechanical behavior of the polymer membrane.

#### 3.2.1. Tensile Tests

In order to investigate the influence of the deposited GO on the PFSA membranes mechanical properties, uniaxial tensile-to-rupture tests were conducted. Figure 4 presents the true stress–strain curves of GO–Nafion^®^ composite membranes compared to their control Nafion^®^ samples under different annealing conditions with the mechanical results summarized in Table 1. The results show that in all cases, the addition of GO to the Nafion^®^ membranes increases % of elongation, making the GO-printed membranes more ductile to unidirectional deformation than their respective control samples without GO.

A significant decrease in the mechanical properties of the membranes without GO was observed as annealing time and temperature increased (Figure 4c,d). While the Nafion^®^ control membranes fractured at elongations of 95% and 98%, their GO-coated counterparts achieved elongations of 248% and 191% respectively. This indicates that the presence of GO stabilizes the membranes mechanically under tension, especially after prolonged annealing at higher temperatures. Moreover, the ductile response of the GO composite membranes during plastic deformation suggests superior toughness in comparison to the bare Nafion^®^.

Additionally, since the strength of PFSA membranes dramatically decreases upon moisture absorption due to the plasticization of the ionomer [80], a sample of pristine Nafion^®^ that was neither thermal treated nor in contact with water was also tested. Figure 5 compares the mechanical behavior of this pristine sample with the samples that were in contact with water during the printing process: both the control sample (Nafion Wet) and the membrane on which surface was printed GO using water as the solvent (Nafion Wet + GO).

In this comparative graphic, a steeper curve is observed on pristine Nafion^®^, indicating that it deforms at a slower rate than the samples that have been in contact with water. This is due to the plasticizing effect of water on Nafion^®^ [81]. However, the sample containing GO not only supports the same amount of deformation as the control sample but exceeds its elongation percentage by more than 30% and its fracture strength by 1.96 MPa. Additionally, a comparison with the dry sample also shows that the presence of GO in the PFSA polymer enhances its ductility without significantly reducing its strength (a decrease of only 0.86 MPa compared to the dry sample). This is a noteworthy mechanical characteristic, especially useful in the humidified environments found in fuel cells and electrolyzers.

The tensile test results suggest that part of the load is transferred from the polymer to the graphene oxide, allowing the composite membrane to withstand higher strain levels and support greater loads before breaking. Yadav et al. [73] and Farooqui et al. [36] reported that GO can improve both the side chains and the backbone of Nafion^®^, resulting in enhanced mechanical and thermal properties. This behavior is expected, as previous studies have shown that the addition of GO significantly improves mechanical properties, primarily due to the hydrogen bonding interactions between GO nanosheets and the Nafion^®^ matrix [32,35].

Hydrogen bond interactions between GO particles and Nafion^®^ due to the OH functional groups present in both materials provides another explanation for the observed mechanical improvements [36]. Appendix A presents a schematic of the chemical structures of Nafion^®^ and layered graphene oxide, highlighting multiple locations where the hydroxyl groups may interact and form hydrogen bonds between the two materials. It shows the hydroxyl groups on Nafion^®^’s sulfonic groups, which are highly hydrophilic and the hydroxyl groups distributed throughout the layers of graphene oxide.

As shown in Figure 2b–d, the bonds between the GO and the Nafion^®^ membrane could help to deflect some of the cracks on the polymer’s surface, absorbing part of the energy and enabling the membrane to withstand higher levels of tensile forces before breaking. The graphene oxide deposited at the surface exhibits a staple effect reinforcing the surface of the membrane. A schematic diagram illustrating this hypothesis is presented in Appendix A.

The improved mechanical performance on the GO–Nafion^®^ samples annealed for extended periods (80 °C for 12 h) and at higher temperatures (110 °C for 3 h), which can be observed on Figure 4c,d, may also be attributed to GO enhancing the thermal stability of the polymer.

#### 3.2.2. Nanomechanical Mapping of the Surface

Since an investigation of the mechanical properties at a microscopic level provides insight into why the composite membrane behaves as it does macroscopically, the quantitative nanomechanical features of the GO islands and Nafion^®^ membrane interface were studied by AFM. Table 2 summarizes the different elastic moduli found locally. In all cases, nanomechanical mapping not only revealed elastic moduli ranging from 7 to 41 GPa on the surface of the GO islands (Figure 6) but also showed that in the proximity of the islands, the elastic moduli of the Nafion^®^ increase. This supports the hypothesis that the connections seen in Figure 2b,c have a mechanical influence in the Nafion^®^ zones adjacent to the graphene oxide.

We can also observe that, consistent with the behavior observed in the tensile tests, the elastic moduli decrease drastically with the increase in both the time and temperature of the thermal treatment, dropping from 260 MPa at 25 °C to 22 MPa at 110 °C. This is expected since the elastic moduli and the proportional limit stress of the membrane decrease as the humidity and temperature increase [82].

Remarkably, while the nanomechanical mapping of samples without graphene oxide revealed the lowest elastic moduli on the samples annealed for 12 h at 80 °C (161 MPa) and the one annealed 3 h at 110 °C (22 MPa), their GO-containing counterparts retained elastic moduli values close to those of the new membrane (357 and 249 MPa, respectively) (Figure 6c,d). This explains the better mechanical performance during tensile tests in terms of the elongation percentage and breaking strength.

#### 3.2.3. Nanoindentation Tests

During the assembling process of PEM fuel cells and electrolyzers, the soft polymer membrane is often clamped between two current collectors, which, if not evenly tightened, produce a risk of perforation [83]. To better understand the mechanical response of the composite membrane to puncture, nanoindentation tests were performed to measure the elastic moduli at specific points.

Six local indentations were performed on the composite membranes. Figure 7 shows a 3D representative image of the indentations that were executed. In the image, the indentations are clearly visible on the Nafion^®^ side (left) but imperceptible on the GO side (right) due to its wrinkle morphology and remarkable elasticity. Nevertheless, the measured values are consistent with the nanomechanical mapping, showing higher values on the GO side than on the Nafion^®^ side (511.46 MPa and 430.75 MPa, respectively).

While elastic moduli values on the Nafion^®^ side are closer to the ones obtained during the nanomechanical mapping (0.300 GPa vs. 0.430 GPa, respectively), the nanoindentation’s GO Young’s modulii are drastically different (0.511 Gpa vs. 33.69 Gpa, respectively). However, when comparing mechanical data between local nanoindentations and full-sample nanomechanical mapping, it should be noticed that due to shallower depth of the indents and the higher data resolution achieved by acquiring a significantly larger number of indents on the same area, nanomechanical mapping is a more reliable technique for characterizing the mechanical properties in samples than nanoindentation [84].

#### 3.2.4. Membrane Swelling

Figure 8 shows the percentage of water swelling for both the control and composite samples. Although all membranes exhibited swelling in the range of 8 to 11%, it is noteworthy that the composite membranes demonstrated slightly lower levels. This may be due to the GO providing some resistance to expansion on the face of the membranes where it was deposited. However, since only one face of the membrane was modified by the GO deposition, it is reasonable to expect that, when fully submerged in water, the membrane would still be able to swell considerably. Despite this, due to the deformation of the membrane in its hydrated state, there is a notable overlap in the error. To account for this, in-plane swelling measurements were performed.

### 3.3. Electrical Characterization

An undesired increase in the electrical conductivity of the composite membrane is not expected since GO is generally described as an electrical insulator due to the disruption of its *sp*^2^ bonding networks [65]. In order to ensure that neither the thermal treatment nor the ultrasonic fabrication process may have functionalized GO, turning it into a more electrically conductive graphene-like material, the sheet resistance of the composite membranes was tested using four-point probe measurement. During the tests, the probes were placed in direct contact on the membrane, and three different points were tested in each membrane to verify electric insulation on the GO-printed surface.

The electrical conductivity (σ) and sheet resistance ($R_{s}$) of the membranes were obtained using the voltage–current (V-I) curves shown on Appendix A. The sheet resistance was calculated using the linear region of the V-I curve and the following relation:(2)$$R_{s}=\frac{V}{I}\cdot k,$$
where *V* and *I* represent the measured voltage and current, respectively, and *k* is the geometric correction factor specific to the four-point probe setup. The electrical conductivity σ was derived from the sheet resistance as
(3)$$\sigma =\frac{1}{R_{s}}\cdot t,$$
where *t* is the thickness of the membrane (0.0127 cm).

The results indicate that the plain Nafion membrane exhibits an electrical conductivity of $1.57\times 10^{-2}\;S/\mathrm{cm}$, while the GO-modified Nafion^®^ membrane shows a significantly lower conductivity of $2.00\times 10^{-4}\;S/\mathrm{cm}$. The corresponding sheet resistances are $5.00\;k\Omega \cdot {\mathrm{cm}}^{2}$ for both cases, reflecting the insulating nature of graphene oxide (GO).

These findings confirm that the deposition of GO on the surface of the polymer does not increase its electrical conductivity. The overall impact is minimal and should not interfere with the adequate performance of the membrane.

### 3.4. Large-Scale Production and Future Perspective

The development of graphene oxide (GO)-reinforced composite membranes presents a cost-effective and scalable approach for large-scale industrial applications. The ultrasonic spray deposition of GO ink onto PFSA membranes enhances their mechanical and hydrophilic properties while maintaining compatibility with PEMWE processes. GO, synthesized via the modified Hummers method, is significantly cheaper than graphene or MWCNTs and uses deionized water as a solvent, making it both economical and easy to handle. Ultrasonic spraying, an energy-efficient and widely adopted technique, directly competes with high-cost processes like chemical vapor deposition (CVD) used for producing graphene or MWCNT nanocomposites. Electrolysis companies already utilize ultrasonic spray machines to fabricate catalyst-coated membranes, ensuring easy adoption due to the existing familiarity among technicians. Its simplicity, reproducibility, and low-energy requirements make ultrasonic spraying ideal for scaling up to mass manufacturing, reinforcing the industrial viability of GO-reinforced membranes.

#### Potential Challenges and Future Work

Despite the promising results, certain challenges and questions remain. While the GO-reinforcement approach significantly enhances the mechanical properties of the Nafion^®^ membranes, further research is required to fully understand how GO affects the membrane electrode assembly (MEA). Future work will focus on assessing the long-term durability of the GO-reinforced PFSA membranes under operational conditions within single-cell real PEM electrolysis systems. Additionally, the impact of graphene oxide on key electrochemical parameters, such as proton conductivity and molecular crossover, will be systematically evaluated. These studies aim to determine the influence of the GO layer on the membrane’s overall electrochemical performance. The findings will be detailed in an upcoming publication, which will also explore the durability of these membranes under extended operational conditions. Finally, other parameters such as the GO concentration in the ink and refining the ultrasonic spray parameters are critical next steps to maximize the performance of the composite membranes. These optimizations aim to achieve reproducible and superior results while maintaining cost effectiveness.

## 4. Conclusions

In this study, a GO solution was ultrasonically sprayed onto Nafion^®^ 115 membranes, and their physical properties were extensively investigated under varying annealing conditions. SEM images showed that a simple thermal treatment at 80 °C for three hours in air effectively promoted interactions between GO and the membrane surface. The GO coverage, which accounted for 35% of the surface, enhanced wettability, reducing the contact angle from 105.2° to 84.4°, while slightly decreasing swelling.

Tensile tests revealed a significant improvement in mechanical properties, with GO-containing membranes exhibiting elongations of 248% and 191%, compared to 95% and 98% for the controls. This demonstrates that GO–Nafion^®^ hydrogen bonds stabilize the membranes under tension, particularly at higher annealing temperatures. Nanomechanical mapping and nanoindentation tests further confirmed increased elastic moduli near the GO–Nafion^®^ interface, suggesting that GO enhances mechanical resilience by suppressing crack propagation.

Importantly, the addition of the GO layer did not alter the membrane’s electrical resistance as evidenced by the measured electrical conductivity of $2.00\times 10^{-4}\;S/\mathrm{cm}$, consistent with the pristine membrane. This work highlights the potential of GO as a means to enhance the mechanical and thermal properties of PEM membranes through a simple, scalable, and automated ultrasonic spray technique.

## Figures and Tables

**Figure 1 nanomaterials-15-00068-f001:**
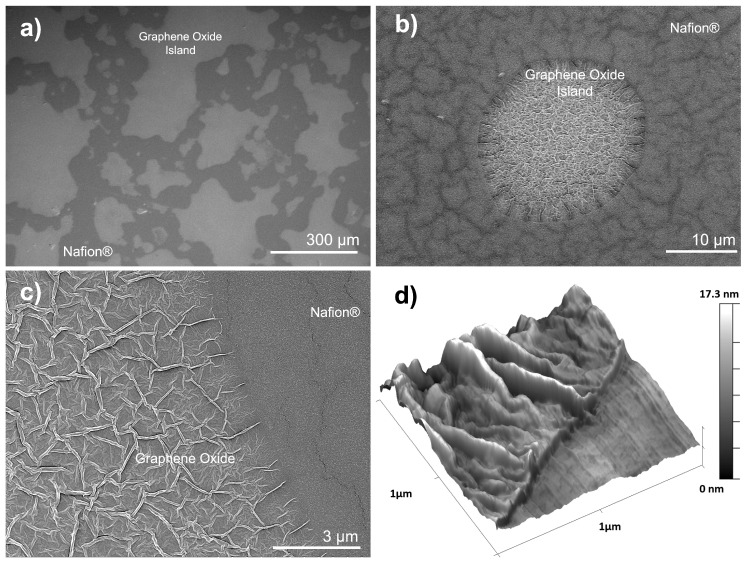
Morphology of the GO islands at different SEM magnifications (**a**–**c**) and AFM image of the graphene oxide–Nafion^®^ interface (**d**).

**Figure 2 nanomaterials-15-00068-f002:**
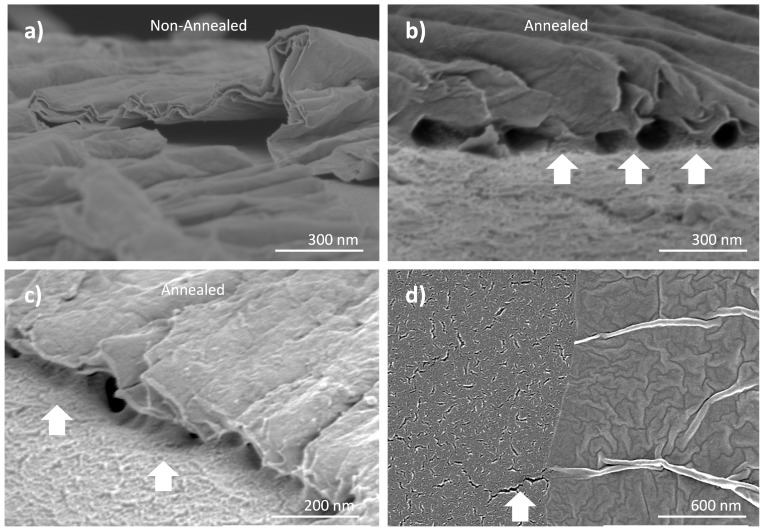
SEM cross-section images of the non-annealed (**a**) and annealed (**b**,**c**) composite membranes. SEM top view of a defect on the polymer side of the composite vanishing upon reaching the GO island (**d**).

**Figure 3 nanomaterials-15-00068-f003:**
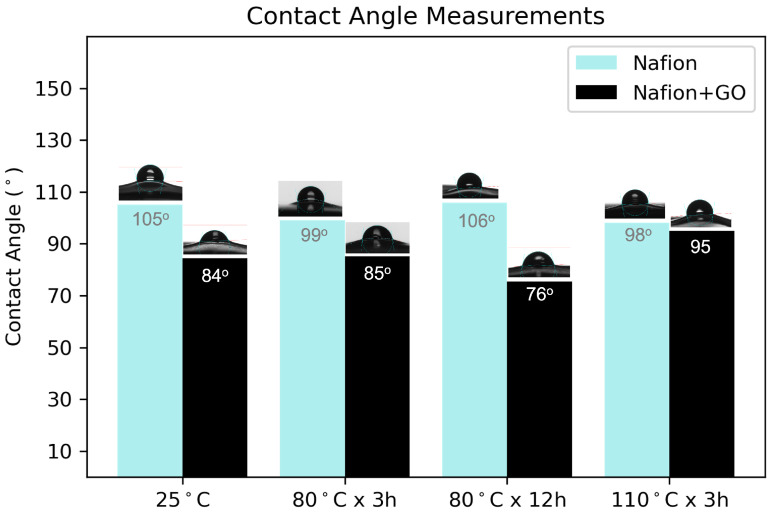
Contact angle measurements on Nafion^®^ membranes with and without ultrasonic sprayed GO deposited on its surface before and after different annealing conditions.

**Figure 4 nanomaterials-15-00068-f004:**
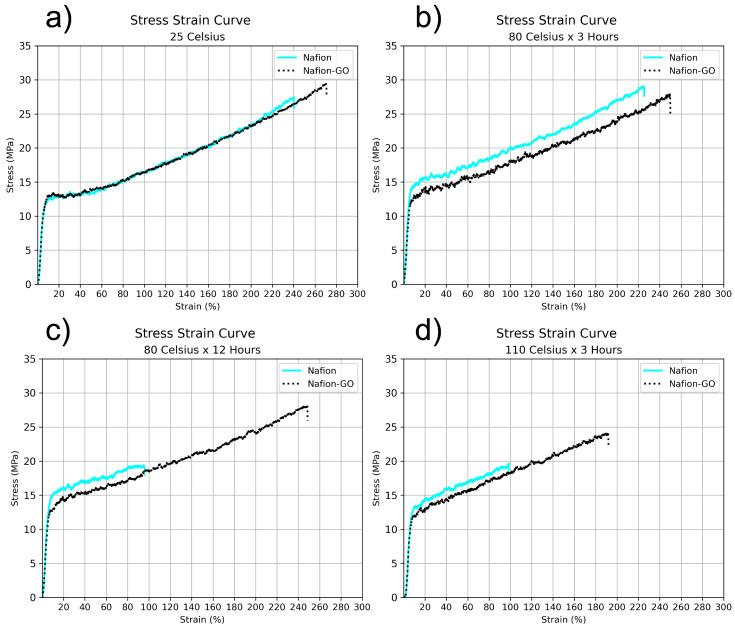
Stress–strain curves of GO composite membranes compared to Nafion^®^ membranes after treatment at different temperatures: (**a**) 25 °C and non-thermal treatment, (**b**) treated at 80 °C for 3 h, (**c**) treated at 80 °C for 12 h and (**d**) treated at 110 °C for 3 h.

**Figure 5 nanomaterials-15-00068-f005:**
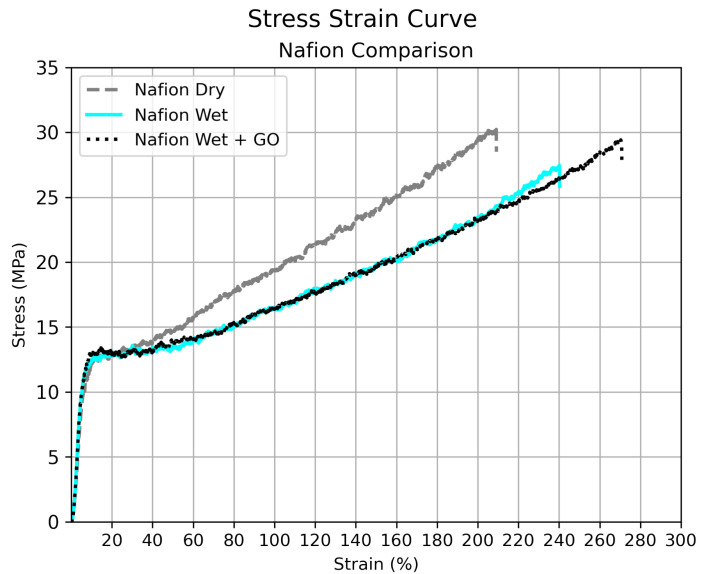
Comparison between Nafion^®^ samples. Pristine membrane (Nafion Dry), control sample in contact with ultrasonic sprayed water (Nafion Wet) and membrane with GO ultrasonically sprayed on its surface (Nafion Wet + GO).

**Figure 6 nanomaterials-15-00068-f006:**
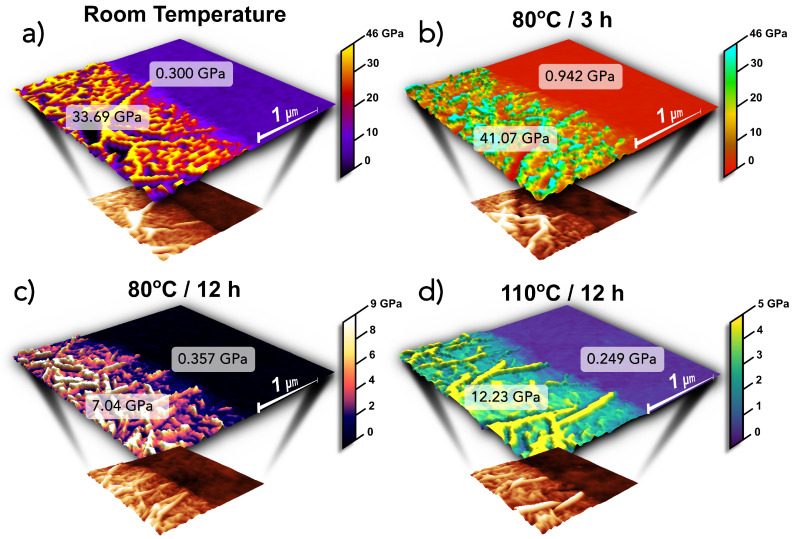
Nanomechanical mapping of composite Nafion^®^-GO membranes after different thermal treatments: (**a**) 25 °C, no thermal treatment; (**b**) treated at 80 °C for 3 h; (**c**) treated at 80 °C for 12 h; and (**d**) treated at 110 °C for 3 h. Despite the various thermal treatments and solvent exposure, the Nafion^®^ side of the membrane maintains Young’s modulus values equal to or greater than those of pristine Nafion^®^, approximately 249 MPa, while the graphene oxide regions exhibit Young’s modulus values that are several orders of magnitude higher than those typically observed in Nafion^®^.

**Figure 7 nanomaterials-15-00068-f007:**
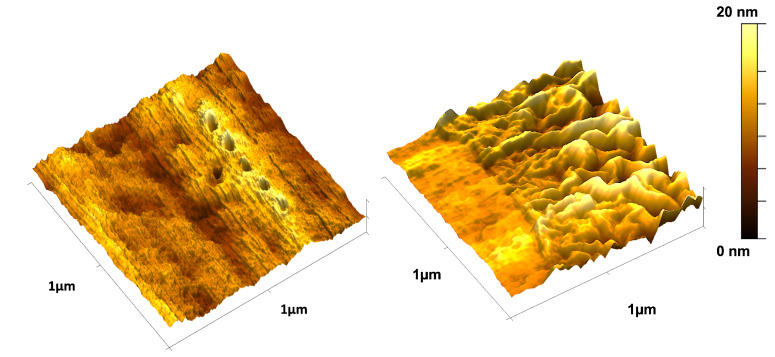
AFM images of indentations performed on the surface of the PEM membranes, showing the Nafion^®^ side (**left**) and the GO side (**right**). The indentations on the Nafion^®^ side are clearly visible, while the wrinkled and elastic behavior of GO renders the indentations imperceptible.

**Figure 8 nanomaterials-15-00068-f008:**
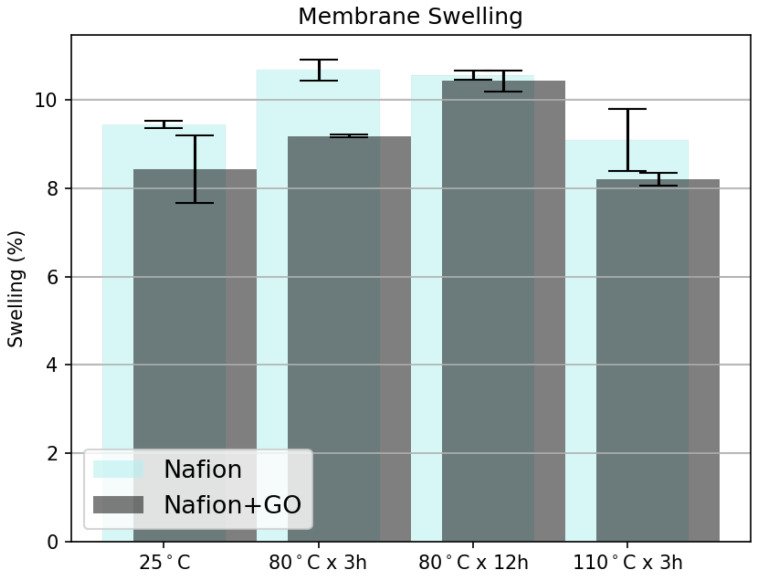
Percentage of membrane water swelling on control samples (Nafion^®^) and composite samples (Nafion^®^ + GO).

**Table 1 nanomaterials-15-00068-t001:** Macro mechanical properties of the composite membranes vs. control Nafion^®^ samples after different thermal treatment conditions.

Thermal Treatment	Breaking Strength (MPa)		Elongation (%)	
	**Nafion**	**GO–Nafion**	**Nafion**	**GO–Nafion**
Non-Treated	27.46	29.42	240.27	270.87
80 °C × 3 h	28.96	27.88	225.46	250.03
80 °C × 12 h	19.42	28.15	95.85	248.70
110 °C × 3 h	19.70	24.07	98.47	191.95

**Table 2 nanomaterials-15-00068-t002:** Nanomechanical mapping values of the control Nafion^®^ samples vs. the composite membranes after different thermal treatments.

Thermal Treatment	Elastic Moduli (GPa)	Elastic Moduli (GPa)	
	**Bare Nafion**	**Nafion Side Interface**	**GO Side Interface**
Non-Treated	0.260	0.300	33.690
80 °C × 3 h	0.089	0.942	41.072
80 °C × 12 h	0.161	0.357	7.040
110 °C × 3 h	0.022	0.249	12.230

## Data Availability

The data is contained within the article.

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
