# Peer review of "Influence of Graphene Oxide on Mechanical and Morphological Properties of Nafion® Membranes"

_nanomaterials, 2025, doi:10.3390/nano15010068_

Round 1
Reviewer 1 Report
Comments and Suggestions for Authors
The authors submitted the article “Influence of Graphene Oxide on Mechanical and Morphological properties of Nafion® Membranes” to Nanomaterials. The authors prepared the Nafion®GO composite for improving the mechanical properties. After reviewed the this manuscript, this manuscript systematically studies the wetting and mechanical properties of composite membranes, and is instructive. The paper may be published in this Journal after minor revisions. Some advice is as followsï¼›
(1) Lines 39-40, on page 2, the authors mentioned “Fabricating composites has been attempted to improve Nafion® mechanical resistance”, I suggest adding specific principles, why can GO improve the above performance, so as to make it easier for readers to understand this kind of material.
(2) Line 172, on page 5, Is GO islands a proper noun, or just a random name, please confirm.
(3) How does the author guarantee that 6 to 7 layers of graphene oxide can be synthesized? Are there SEM images of GO synthesis.
(4) Lines 234-237, on page 7. What does the contact Angle have to do with anything other than the bonding force.
(5) What is the proton conductivity of the composite membrane (Nafion®GO). And what the electrical conductivity of the GO membrane in Figure S7?
(6) The conclusion is too long. It is suggested that it be shortened and refined.
(7) I suggest that the authors add additional research work for 2023 and 2024.
Author Response
Dear Reviewer 1, we sincerely thank you for your valuable comments and suggestions.
"Please refer to the attachment for the answers"

Reviewer 2 Report
Comments and Suggestions for Authors
This manuscript presents a comprehensive study on the development and characterization of graphene oxide (GO)-reinforced Nafion composite membranes for proton exchange membrane (PEM) applications. The authors employ a systematic approach, integrating material synthesis, morphological characterization, and mechanical testing to evaluate the performance of the composite membranes. The study is well-structured, and the findings make a significant contribution to the growing body of knowledge on PEM technologies. However, several areas could be improved to enhance its clarity, scientific rigor, and overall impact:
-
A more detailed comparison with other reinforcement strategies would strengthen the manuscript, particularly by highlighting how GO outperforms alternatives such as multiwalled carbon nanotubes or graphene in areas like mechanical stability, hydrophilicity, and electrical insulation.
-
Including a brief discussion on the feasibility of the GO-reinforcement approach for large-scale production, specifically addressing cost implications, energy requirements, and potential challenges, would enhance the industrial relevance of the study.
-
Assessment of the long-term durability of the GO-reinforced Nafion membranes under operational conditions, such as those in electrolysis applications (e.g., 10.1021/acscatal.3c01439; 10.1126/science.aay4217), would provide valuable insights into their practical usability.
-
Incorporating data on the influence of GO on the molecular crossover properties and proton conductivity of Nafion would add depth to the analysis and strengthen the study’s contributions (e.g., 10.1039/d1ee01664d; 10.1016/j.jcat.2020.03.013).
Author Response
Dear Reviewer 2, we sincerely thank you for your valuable comments and suggestions.
"Please refer to the attachment for the answers"
